# The Immune Response of Cutaneous Basosquamous- and Squamous-Cell Carcinoma Associated with Sun Exposure

Anamaria Grigore [1,*] , Ana-Maria Oproiu [1,2,*] , Ioana Iancu [3] and Ioan-Petre Florescu [1]

1   Plastic Surgery Department, "Carol Davila" University of Medicine and Pharmacy,
    020021 Bucharest, Romania
2   Plastic Surgery Department, Emergency University Hospital, 050098 Bucharest, Romania
3   Manchester Centre for Clinical Neuroscience, Manchester M6 8HD, UK; ioana.iancu@nca.nhs.uk
*   Correspondence: ana.grigore080892@gmail.com (A.G.); anamaria.oproiu@umfcd.ro (A.-M.O.)

**Abstract:** In recent years, there has been an observed increase in the frequency of cutaneous carcinoma, which correlates with sun exposure. This study aims to explore the variances of tumor characteristics and immune response markers among patients diagnosed with cutaneous squamous-cell carcinoma (SCC) and basosquamous-cell carcinoma (BSC) with varying levels of sun exposure. The objective is to elucidate the potential influence of sun exposure on tumor progression and immune response in these types of carcinomas. We conducted a retrospective observational study that included 132 patients diagnosed with SCC and BSC. Participants were separated into high- and low-sun exposure groups. Tumor characteristics and immune response markers, including lymphocyte percentage (LY%), neutrophil-to-lymphocyte ratio (NLR), and lymphocyte-to-monocyte ratio (LMR), were assessed using the Mann–Whitney U test. Our findings revealed the interplay between sun exposure, inflammation, aging, and immune response. In 80% of cases, it was found that individuals had high sun exposure throughout their lifetime. Patients in the high sun exposure category had a significantly higher LY% than those with low sun exposure (24.22 ± 7.64 vs. 20.71 ± 8.10, $p = 0.041$). Also, the NLR was lower in patients with high sun exposure (3.08 ± 1.47 vs. 3.94 ± 2.43, $p = 0.023$). Regarding inflammatory markers, the erythrocyte sedimentation rate (ESR), LY%, NLR, and LMR showed significant differences between the two groups. Patients who were diagnosed with SCC had higher ESR values ($p = 0.041$), higher LY% ($p = 0.037$), higher NLR ($p = 0.041$), and lower LMR ($p = 0.025$). This study provides evidence supporting distinct tumor characteristics and immune response patterns in patients diagnosed with SCC and BSC with a high sun exposure history. These findings imply that sun exposure may contribute to tumor progression and influence the immune response in individuals with SCC and BSC.

**Keywords:** squamous-cell carcinoma; basosquamous-cell carcinoma; sun exposure; immune response

## 1. Introduction

Squamous-cell carcinoma is one of the most common skin cancers, ranking fifth in the list of health system costs caused by cancers. With a steadily increasing incidence, it is estimated that in the US alone, more than 1 million squamous-cell skin cancers are diagnosed per year, and approximately 2500 deaths occur as a result [1]. It is a type of cancer that grows very fast, often metastases, and has a high risk of recurrence [2]. It can appear on healthy tissue and pre-existing lesions, such as actinic keratosis. Cutaneous squamous-cell carcinoma (SCC) has a predisposition to occur in aged people, especially in areas exposed to the sun, such as the head and neck. A possible theory explaining the increased incidence supports the role of ultraviolet radiation, specifically the damage to the stratospheric ozone that leads to much stronger solar radiation [3].

Controversially, basosquamous-cell carcinoma (BSC) is a borderline tumor between basal-cell carcinoma and squamous-cell carcinoma, exhibiting common features of both

types of skin cancers. Histologically, it presents a combination of different proportions of basal- and squamous-cell carcinoma and a transition zone between them [4]. BSC has been a controversial carcinoma for many years, with the latest definition used by the World Health Organization for BSC being that "Basosquamous carcinoma is a term used to describe basal cell carcinomas that are associated with squamous differentiation" [5].

The particularity of this type of cancer is its aggressive behavior, similar to squamous-cell carcinoma, and the absence of a targeted treatment protocol [6]. This type of carcinoma is poorly described in the literature, with a low reported incidence ranging from 1.7% to 2.7% [7]. A prospective study conducted by Gualdi et al. between 2012 and 2015, which included 6042 patients, revealed a higher incidence of 4.8% for BSC [8]. Previous studies reported incidences of 1.2% by Schuller et al., 1.4% by Martin et al., and 2.7% by Bowman et al. [9,10].

Both SCC and BSC have been associated with sun exposure, necessitating a better understanding of the sun's exposure effects. Ultraviolet radiation serves as both an initiator and a promoter in cutaneous carcinoma, with sun exposure causing abnormalities in oxygen reactions, deoxyribonucleic acid (DNA) damage, and lipid and protein damage, and triggering inflammatory and immunosuppressive processes [11]. Cumulative oxidative stress and DNA damage also contribute to hematopoietic stem cell senescence, which is associated with inflammation, aging, and immune cell abnormalities [12–14].

Based on these observations, SCC and BSC could result from the interplay between sun exposure, skin aging, inflammation, and immunosenescence. This interplay may lead to an insufficient defense system and promote carcinogenesis. This paper aims to investigate the tumor characteristics and immune response markers between SCC and BSC in patients with low and high sun exposure. The purpose of this study is to elucidate the potential impact of sun exposure on tumor progression and immune response in SCC and BSC.

## 2. Methods

This study is a retrospective observational single-center study conducted at the Plastic Surgery Clinic of Emergency University Hospital in Bucharest, focusing on patients diagnosed with SCC and BSC through histopathological examination. A total of 132 patients diagnosed with SCC and BSC were included in this observational study. Preoperative assessment included a complete blood count. Blood samples were analyzed in a certified laboratory, following standard procedures and protocols by current international and national guidelines. The neutrophil-to-lymphocyte ratio (NLR), lymphocyte-to-monocyte ratio (LMR), and lymphocyte percentage (LY%) were estimated as immune response markers. Also, the erythrocyte sedimentation rate (ESR) was used to evaluate the inflammatory status. Based on participants' self-reported sun exposure history and tumor location, they were divided into two groups according to sun exposure level: high and low. Sun exposure levels were defined based on the self-reported average number of hours spent in the sun without appropriate sunscreen protection. The immune and inflammatory status, measured through LY%, NLR, LMR, and ESR, was analyzed using the Mann–Whitney U test. A $p$-value less than 0.05 was considered to be statistically significant. Numerical values were reported as mean $\pm$ standard deviation (SD) to denote the central tendency and variability of the data. The study was conducted with the approval of the Ethics Committee of Emergency University Hospital, and all participants provided informed consent during their hospitalization.

## 3. Results

Out of 132 participants, 53% were male and 47% female. Approximately 67% of patients were diagnosed with SCC, and 33% were diagnosed with BSC. A predominance was observed among patients aged 70 to 79 years old (37.90%) and 80 to 89 years old (30.30%). The smallest percentage of patients was seen in the group below 60 years of age, with only 8.4% of patients belonging to this group. Exposure to solar radiation was significant, with 80% of patients showing high lifetime exposure (Figure 1).

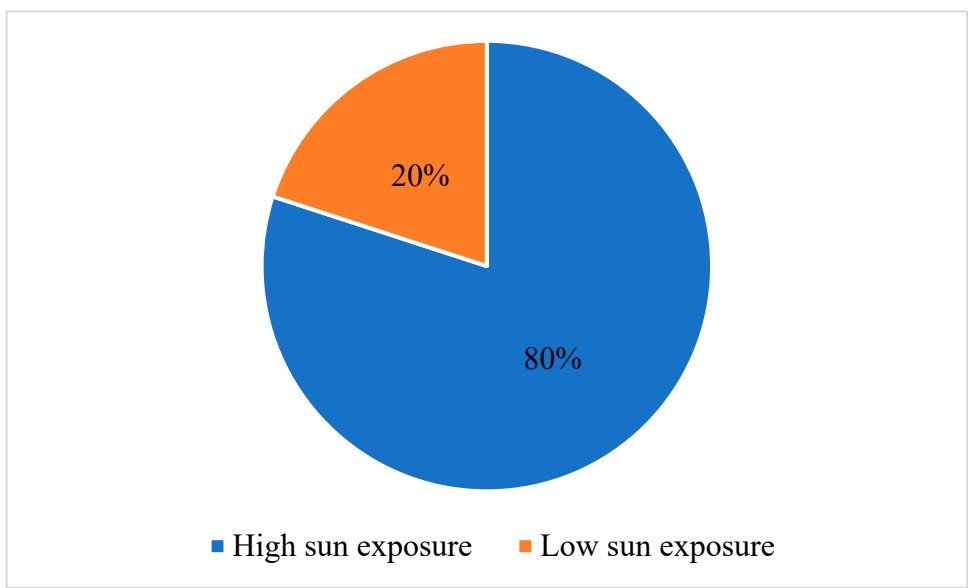

**Figure 1.** Distribution of patients diagnosed with cutaneous squamous-cell carcinoma (SCC) and basosquamous-cell carcinoma (BSC) according to sun exposure.

Another significant statistical finding was the difference between the LY% in the high and low sun exposure groups (Table 1). Patients with high sun exposure had a higher LY% than those with low sun exposure (24.22 ± 7.64, versus 20.71 ± 8.10, $p$ = 0.041). Patients with low sun exposure had higher NLR and lower LY% compared to those with higher sun exposure (3.94 ± 2.43 versus 3.08 ± 1.47, $p$ = 0.023).

**Table 1.** Biomarkers in patients with high versus low sun exposure.

| Variable | High Sun Exposure (n = 106, 80%) ± SD | Low Sun Exposure (n = 26, 20%) ± SD | t | df | p |
|---|---|---|---|---|---|
| LY% | 24.22 ± 7.64 | 20.71 ± 8.10 | 2.067 | 130 | 0.041 |
| NLR | 3.08 ± 1.47 | 3.94 ± 2.43 | −2.309 | 130 | 0.023 |

The data were also analyzed comparing the two tumor groups—patients with SCC (n = 88, 66.66%) versus patients with BSC (n = 44, 33.33%).

Regarding participants' age, the difference between the two groups was not statistically significant (Table 2).

**Table 2.** Biomarkers in patients with cutaneous squamous-cell carcinoma (SCC) versus patients with basosquamous-cell carcinoma (BSC).

| Variable | SCC Group (n = 88) ± SD | BSC Group (n = 44) ± SD | t | df | p |
|---|---|---|---|---|---|
| Age (years) | 76.14 ± 12.50 | 74.86 ± 9.53 | 0.594 | 130 | 0.554 |
| ESR (mm/h) | 27.54 ± 18.48 | 20.65 ± 16.92 | 2.059 | 130 | 0.041 |
| LY% | 25.54 ± 7.90 | 22.53 ± 7.38 | −2.108 | 130 | 0.037 |
| NLR | 3.47 ± 1.93 | 2.81 ± 1.12 | 2.062 | 130 | 0.041 |
| LMR | 2.96 ± 1.45 | 3.67 ± 2.10 | −2.263 | 130 | 0.025 |

However, the body's immune and inflammatory response measured by ESR, LY%, NLR, and LMR showed significant differences between the two groups.

Patients who were diagnosed with SCC had higher ESR values ($p$ = 0.041), LY% ($p$ = 0.037), and NLR ($p$ = 0.041), and lower LMR ($p$ = 0.025) (Table 2, Figures 2 and 3).

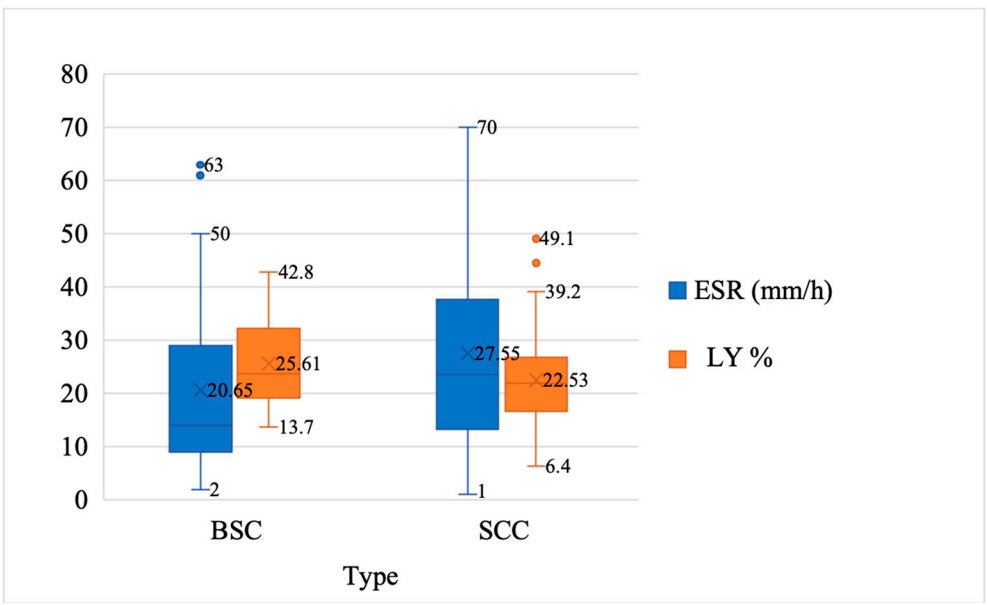

**Figure 2.** Medium, minimum, and maximum value of erythrocyte sedimentation rate (ESR) and lymphocyte percentage (LY%) among patients with basosquamous-cell carcinoma (BSC) and cutaneous squamous-cell carcinoma (SCC).

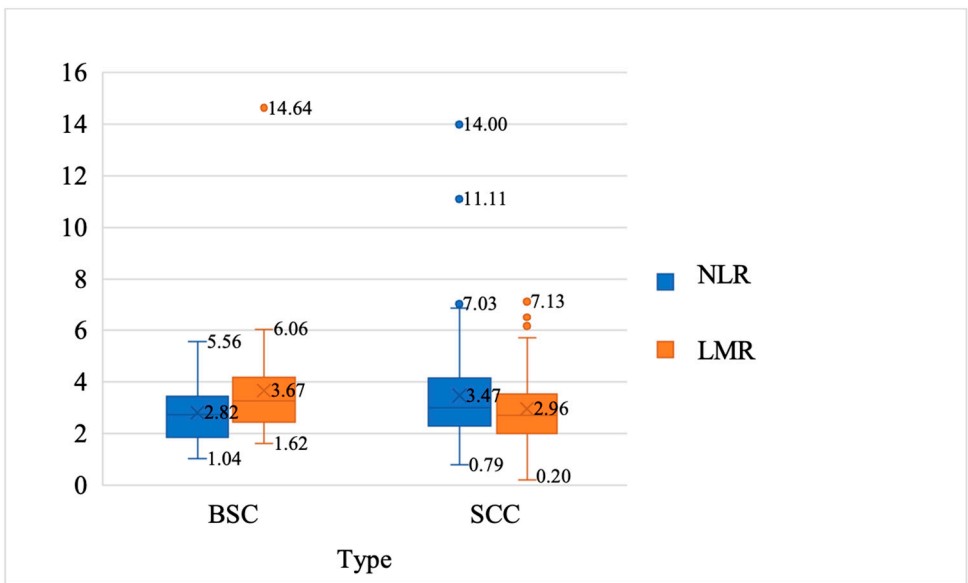

**Figure 3.** Medium, minimum, and maximum value of neutrophil-to-lymphocyte ratio (NLR) and lymphocyte-to-monocyte ratio (LMR) among patients with BSC and SCC.

Furthermore, the statistical analysis showed a higher level of ESR in the rural patient group, thus resulting in a higher prevalence of sun exposure than in the urban group. This also correlated with a larger tumor size, a difference that was statistically significant. In absolute values, clear differences could be observed between $22.86 \pm 13.24$ mm in urban areas and $29.50 \pm 21.96$ mm in rural areas.

## 4. Discussion

Although the available data classify SCC as the second most common skin cancer, with a constantly increasing worldwide prevalence [15], and BSC as a rare cancer which does not have a standardized approved therapeutic guidelines for the management [16], our experience has shown that BSC is not rare. Therefore, it requires a customized treatment

protocol. It should be treated as an individual entity that can, in certain cases, be more aggressive than SCC.

An estimated 2.75 million cutaneous squamous-cell cancers are diagnosed annually, with the increased incidence associated with damage to the ozone layer and increasingly aggressive ultraviolet radiation. The development of squamous-cell carcinoma of the skin is linked to chronic and early exposure to the sun, with approximately 90% of skin carcinoma cases (excluding melanoma) attributed to sun exposure [17].

The theory that inflammation causes cancer has been around since 1828. In 1863, Virchow was the first to observe the presence of leukocytes at the tumor site, thus linking the processes of inflammation and carcinogenesis [18–21]. As a result, numerous studies have explored the role of inflammation in cancer initiation, progression, and spread [22].

Focused on cutaneous carcinoma, a history of sun exposure leads to local inflammation, which can promote carcinogenesis, and many studies have paid attention to that. Little et al., in a large prospective cohort, demonstrated that cumulative ultraviolet radiation exposure, especially on anatomical sites like the face and head, predisposes individuals to skin cancer [23]. Martincorena et al. utilized genomic sequencing to show that prolonged sun exposure leads to mutations in keratinocytes long before the appearance of precancerous lesions and squamous skin carcinomas [24]. Thus, at the epithelial level, DNA changes and prolonged exposure to the sun are associated with the development and progression of certain cancers [25,26].

In this study, we observed an increased incidence of SCC and BSC in aged individuals with a history of sun exposure. Compared with the literature reporting that 82–97% of BSC cases occur in sun-exposed areas of the head and neck, with a prevalence in aged individuals [16], our study confirmed the prevalence for sun-exposed areas and older adults. Concerns about aging date back to 1939, when Walford introduced the theory of immunological aging. In 1969, this led to the development of the immunosenescence concept, which involves abnormalities in immune cellular activities due to aging [27].

Immunosenescence, a consequence of aging, manifests as a gradual decline in immune cell functions and body defense processes. Although aging is a physiological process, it impacts all cell functions and may lead to debilitation, disease, and death. The aging process results in the overstimulation of the immune system, leading to systemic inflammation. There is an interplay between aging, inflammation, and immunosenescence [28].

Hematopoietic cells like neutrophils and monocytes, released into the blood to defend against inflammation and pathogens, exhibit weakened functions during aging, contributing to the immunosenescence process. Data show that neutrophils are involved in immune suppression and carcinogenesis through interactions between cancer cells and immune cell blood [29]. Lymphocytes, responsible for the innate immune system, are vital for maintaining human health [30]. LY%, NLR, and LMR can predict the immune status, antitumor defense capacity, and treatment response [31].

Our theory is based on the correlation between sun exposure, aging, inflammation, and cancer. Aging decreases the immune defense, the cumulative radiation effects are higher in the elderly, prolonged sun exposure generates local inflammation, and all of these can initiate and promote cutaneous carcinoma. The purpose of this paper was to evaluate if SCC and BSC associated with sun exposure and inflammation could lead to changes of peripheral blood immune cells.

We measured immune and inflammatory status using peripheral blood. Our study assessed the immune body response to inflammation, reflected in ESR, LY%, NLR, and LMR values. There were slightly higher values of ESR in patients with SCC compared to those in patients with BSC ($27.55 \pm 18.48$ mm/h, versus $20.65 \pm 16.92$ mm/h, $p = 0.041$). Also, patients who were diagnosed with SCC had higher LY% ($p = 0.037$) and NLR ($p = 0.041$), and lower LMR ($p = 0.025$) than patients who were diagnosed with BSC. The fact that 80% of our patients present high lifetime sun exposure supports the theory of interplay between sun exposure and immune interaction. Our findings revealed that immune modifications correlated with sun exposure and aging.

Given that surgery remains the primary treatment for cutaneous carcinoma such as BSC and SCC and because, despite medical progress, the incidence of these cancers is increasing, understanding the immune response in carcinogenesis is crucial for decreasing the incidence and improving other treatment lines such as immunotherapy. The measurement of NLR, MLR, LY%, and ESR from peripheral blood is simple and cheap and can provide information about the body's immune status.

## 5. Conclusions

In this paper, we propose the idea that cumulative solar radiation features play a significant role in the initiation and promotion of carcinogenesis in SCC and BSC. Additionally, there is an interplay between aging, inflammation, immune response, and carcinogenesis.

This study underlines the importance of immune system knowledge in preventing and decreasing cancer and developing new lines of immunotherapy.

Furthermore, our study's ratio between SCC and BSC emphasizes that cases of BSC are much more frequent than previously described in the literature. This underscores the need for a more comprehensive approach to understanding and addressing BSC.

**Author Contributions:** A.G., conceptualization and writing (original draft preparation); A.-M.O., methodology and writing (review and editing); I.I., statistical analysis; I.-P.F., supervision. All authors have read and agreed to the published version of the manuscript.

**Funding:** This research received no external funding.

**Institutional Review Board Statement:** This study ran in accordance with the ethical standards established in the Helsinki Declaration of 1964 and its subsequent amendments. It has the ethical approval of Emergency University Hospital from Bucharest for publishing, the ethical registration number is 23356.

**Informed Consent Statement:** All participants provided informed consent during their hospitalization.

**Data Availability Statement:** The data presented in this study is available on request from the corresponding author.

**Conflicts of Interest:** The authors have no conflicts of interest or financial ties to disclose.

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
