# Peer review of "The Immune Response of Cutaneous Basosquamous- and Squamous-Cell Carcinoma Associated with Sun Exposure"

_curroncol, doi:10.3390/curroncol31050185_

Round 1
Reviewer 1 Report
Comments and Suggestions for Authors
Brief Summary
The manuscript entitled “The immune response of cutaneous basosquamous and squamous cell carcinoma associated with sun exposure” was submitted to Current Oncology. It focuses on the nonmelanoma skin cancers cutaneous squamous cell carcinoma (SCC) and basosquamous cell carcinoma (BSC). The authors report on differences in immune responses observed in patients with varying levels of sun exposure. The manuscript must be improved before acceptance for publication.
General comments
· · The manuscript is mostly clear, although some grammar revisions may be required. Overall, it is relevant; however, it requires changes for a better understanding from the readers.
· If possible, the cited references could be more recent.
· In general, the manuscript is scientifically sound; however, more details regarding patient information, and methods should be included.
· Presented Figures are generally appropriate but could be displayed differently. For instance, the four Figures could be combined in a single panel. Figures and Tables should include information regarding abbreviations and statistical analysis. A Table and/or Figures displaying data for high sun exposure vs low sun exposure should be included since it is the focus of the manuscript.
· The conclusions are somewhat difficult to interpret considering the results and discussion. It is referred a linkage between sun exposure, aging, inflammation and immune response; however, there is no explanation on how the results show this interplay.
· In the ethical statements it should be included the approval by an ethical committee.
Specific comments
· · After defining the abbreviations, please use them through the text.
· · I believe you could provide a summary of available treatment options and respective success in both SCC and BSC so that the readers understand the mortality and morbidity.
· · In Introduction, line 42, according to the paper, the longer life expectancy can also justify the increasing incidence. The authors could include this information since aging is also focused on the manuscript.
· · In Introduction, lines 48-53, the authors describe the reported values of incidence for BSC. From this information, the incidence rate is not consensual. Still, is it considered rare compared to other skin cancers? Also, in the Conclusion, the authors state “Furthermore, our study's ratio between SCC and BSC emphasizes that cases of BSC 192 are much more frequent than previously described in the literature.” Could the authors give a percentage and compare with the literature reports?
· · In lines 59-61, the authors should provide a reference(s).
· · Between lines 59-68, the information seems repetitive. I would ask the authors to revise the text in order to avoid this.
· · In the Methods, the authors should indicate how high and low exposure groups were defined. What parameters did the authors considered when defining low and high? Also, in lines 76-77, please define what were the “appropriate statistical tests”.
· · The focus of the paper seems to be sunlight exposure. Nevertheless, the authors do not describe any significant data on this subject. Only briefly state differences among males/females and LY% and NLR. No Tables and no Figures are presented for sun exposure data. Therefore, in the Results section, between lines 84-103, it would be relevant to include a summary Table and/or Figure with the reported patients’ information and obtained results. This will help the readers better understand which parameters were assessed (e.g.: male/female, age, living area, sun exposure, tumor localization, etc.), as well as to interpret low sun exposure versus high sun exposure. Moreover, in the text, also indicate in which Table or Figure the information is.
· · In Table I and in all Figures, please include, in the footnote/in the caption, information regarding statistical analysis and definition of abbreviations. Did the authors perform normality test of populations? You should also indicate if it is mean +/- SD or other measure.
· · If possible, it would be beneficial to place Figures 1,2,3 and 4 in a single panel.
· In line 137, what kind of connection is established between inflammation and carcinogenesis? Is carcinogenesis potentially exacerbated by inflammation or is inflammation a response to carcinogenesis?
· · In lines 151-153 and 159-162, the cited reference [18] does not have the information written.
· · The relevance/impact of immunosenescence in the development or modulation of skin cancer, namely SCC and BSC, is not clearly defined. Is the cancer progression worsened by immunosenescence or does cancer contributes to immunosenescence?
· · In line 175, surgery is introduced as a treatment for both cancers. The authors should include information regarding treatment options in the Introduction section. This way, the readers will better understand the existing challenges in therapeutic management and the need to comprehend the disease mechanism more deeply.
· · In lines 177-178, could the authors elaborate on these aspects? How does the obtained knowledge may provide advances in terms of prevention and treatment? This is important since one of the journal’s aims is to publish articles that have potential for “application to the current or future practice of cancer medicine”.
· · The authors state that carcinogenesis and aging lead to abnormal immune cell responses; however, the authors should indicate more clearly if sun exposure aggravates or increases the incidence of these skin cancers in older people. Moreover, a significant difference in immunological parameters between SCC and BSC was obtained. Could the authors explain the implications of these findings? How does radiation exposure/age/type of cancer correlate with each other? How different were the ages of patients with SCC and BSC, and could this different age gap affect the obtained data in terms of immune response?
Comments on the Quality of English LanguageEnglish language requires minor revision.
Author Response
Thanks for your feedback. It is very helpful to improve my work. I have found justifiable your comments and I hope the new version will fit with your requires.

Reviewer 2 Report
Comments and Suggestions for Authors
Brief summary
The paper focuses the attention on a very interesting topic
The structure of this paper is clear.
English level is good.
No ethical problems are found in this study
However, I would like to make some suggestions.
General concept comments
I suggest to classify squamous cell carcinoma according to the last WHO classification of skin tumors
I also suggest to add the TNM stadiation sysyem to your cases according to AJCC VIII ed.
Moreover, the discussion part should be add with more literature references.
Comments on the Quality of English Languageenglish level is good
Author Response
Thanks for your feedback. It helps me to improve my work and I hope that the new version will fit with your requires.

Round 2
Reviewer 1 Report
Comments and Suggestions for Authors
Overall, the authors have addressed the reviwer's questions and have provided adequate changes to the manuscript. Nevertheless, to further improve the quality of the reported study, the authors should address the following comments:
In lines 53-54, please use the abbreviation for basosquamous cell carcinoma.
In line 63, instead of “Based on this hypothesis” I would suggest “Based on these observations”.
In line 67, please change the sentence “…with a history of sun exposure.” to “…with low and high sun exposure”.
In lines 79-80, please change the sentence “Based on participants' self-reported exposure history and tumor location, they were divided into two groups based on their level of sun exposure: high and low” to “Based on participants' self-reported sun exposure history and tumor location, they were divided into two groups according to sun exposure level: high and low”.
In line 82, please eliminate “through thorough clerking”.
In lines 84-86, please remove the sentence “The statistical significance of differences in immune and inflammatory status among the different groups, assessed by LY%, NLR, LMR, 85 and ESR, was evaluated using the Mann-Whitney U test”. This information is already given in the sentence that precedes it.
In Figure 1, please remove the additional caption that is placed within the graph. It is not necessary.
Table II and Figure 1 have associated data and should appear closer to each other in the manuscript.
I have noticed that the authors only introduce the urban and rural living areas in the results section. It would be beneficial to add some information in the Methods section.
Information in lines 105-107 is repeated in lines 125-129. Please correct.
The abbreviation “CSC” appears twice in the manuscript. Please correct.
The authors have added the following information in the Discussion: “Also, there is a demonstrated relationship between abnormalities of peripheral blood immune cells and tumor environment.” Throughout the manuscript, tumor microenvironment is not referred to. The authors should provide a reference for this information.
The information in lines 196-201 should appear in the beginning of the Discussion section.
In the last sentence of the Conclusion section, the authors wrote “basosquamous cell carcinoma”. The abbreviation BSC should be used instead.
Comments on the Quality of English Language
Minor English language corrections are recommended.
Author Response
I took into account all your advices and I made all the changes that you requested for. Your advices were very useful to improve my work. Thank you for your involvement!

Reviewer 2 Report
Comments and Suggestions for Authors
Suggestions have been considered and the current version is more complete than the old one
Author Response
All your comments were very useful to improve my work.
